# Impact of Different Amaranth Particle Sizes Addition Level on Wheat Flour Dough Rheology and Bread Features

**DOI:** 10.3390/foods10071539

**Published:** 2021-07-03

**Authors:** Ionica Coțovanu, Silvia Mironeasa

**Affiliations:** Faculty of Food Engineering, Stefan cel Mare University of Suceava, 13 Universitatii Street, 720229 Suceava, Romania; ionica.cotovanu@usm.ro

**Keywords:** composite flour, amaranth, particle size, bread, dough, rheology

## Abstract

The objective of this investigation was to evaluate the effects generated by amaranth flour (AF)—of different particle sizes (PS) added to white wheat flour from 0% to 20%—on the proximate composition, dough rheological behavior, and bread technological parameters. The reduction of particle size led to an increased hydration capacity of the wheat–amaranth composite flour, while bulk density decreased. Increasing the amount of AF and decreasing the PS led to a significant increase in protein, lipids, and ash contents, while the moisture and carbohydrates of the composite flour decreased. Increasing AF addition led to an increase in dough tenacity and a decrease in dough extensibility, while the PS had an irregular trend. The large particle size, at 15% and 20% levels of AF in wheat flour, increased significantly (*p* < 0.001) the dough tenacity and hardness, bread firmness, but decreased bread volume, porosity, and elasticity, while medium and small particles at 5–15% addition levels improved porosity and elasticity of the composite bread. Significant correlations (*p* < 0.05) were found between proximate composition, dough rheological characteristics, and bread quality for the wheat–amaranth composite flours. The results of this study are an important basis for the development of innovative wheat–amaranth bread recipes.

## 1. Introduction

Unrefined flour-based products demand new raw materials to improve bread’s nutritional quality. Recently, consumers have become more aware of the importance of food quality [1]. A good alternative for improving the quality of refined wheat flour is supplementation with unconventional raw materials, rich in nutritional substances. Recent studies demonstrated that amaranth provides many nutritional and health-promoting values that enhance bread’s nutritional value [2,3,4]. Amaranth is a rich source of high-quality proteins with a complex content of amino acids (lysine, threonine, and methionine), high-quality oil, vitamins, minerals, and bioactive compounds [5,6]. Miranda Ramos et al. [3] demonstrated the nutritional effect of amaranth flours when they were added to wheat flour by enhancing the protein and dietary fiber content of bread. The enriched bread also had higher levels of lipids, minerals, and myo-inositol phosphates, which could supply a high percentage of Fe and Zn of the daily human requirement. Pseudocereal amaranth is a gluten-free grain and its addition in composite flour in the bread-making process is a challenge. Some authors demonstrated that less than 20% amaranth replacement positively influenced bread physical parameters, especially the loaf volume, due to the reduction of gluten content in composite flour [7,8]; however, Burisova et al. [9] and Miranda-Ramos et al. [3] showed that amaranth content higher than 20% decreased the content of gluten and had a negative effect on the dough quality (adhesiveness) and bread (low volume and flavor). In recent years, a considerable number of researchers have also focused on highlighting the technological benefits that the addition of pseudocereals brings to bread when innovative manufacturing techniques are employed (e.g., reduction of particle size and use of sourdough) and adequate proportions of pseudocereals are used [3,4]. Particle size reduction is obtained through seed grinding and is an important point of view in obtaining highly nutritious flours, which positively influence the dough technological and rheological parameters, and the bread quality [10]. Previous investigations reported that dough rheological properties can be significantly influenced by pseudocereal particle fractions [11,12,13]. Observing the dough behavior on the technological phases explains how it might be affected in the process. Dough behavior represents an important indicator of the final product’s sensorial qualities [14]. The alveograph can be a useful device for the examination of the rheological parameters which characterize the extensibility of dough [15]. Dough rheology (extensibility, elasticity, and resistance to deformation) is very important in the technological process [16]. Texture profile analysis is an essential instrument for the evaluation of intermediate (dough) and final baked products. Dough consistency is a determining factor for bread quality [17]. Among the bread quality parameters, the crumb texture, volume, porosity, and elasticity are all important criteria in a sensorial analysis.

To the best of our knowledge, no studies have been conducted to compare the effect of amaranth flour at different addition levels and particle sizes on wheat dough rheological properties by using the Alveograph device and on bread quality characteristics.

This research aims to study the influence of amaranth at three different particle sizes (large, medium, and small) in wheat flour of 650 type for bread making at five levels (0%, 5%, 10%, 15%, and 20%) of substitution on Alveograph dough rheological properties, dough texture, and bread quality related to the physicochemical characteristics. The relationships between the proximate composition of wheat-amaranth composite flours, dough rheological properties, and bread features were also analyzed by using the principal component analysis method. 

## 2. Materials and Methods

### 2.1. Basic Materials

The amaranth grains were purchased from S.C. SOLARIS PLANT S.R.L. (Ilfov, Romania). According to the standard analysis methods, amaranth seeds indicated the following characteristics: moisture content (14%), determined by a gravimetric method (ICC 110/1); the fat content (8.00%), determined by extraction with ether through the Soxhlet method (VELP Scientifica, Usmate Velate (MB), Italy) (ICC 104/1); protein content (17.00%), which was measured with a rapid Kjeldahl apparatus using digestion followed by steam distillation (VELP SCIENTIFICA, Usmate Velate (MB), Italy) and the values were expressed as percent protein (N × 5.53) (ICC 105/2); and ash content (3.00%) was determined in a muffle by incineration at 900 °C (ICC 106/1), as a percentage of the dried substances and total carbohydrates content 58.00%, calculated by difference {100- (moisture+fat+protein+ash)} [13]. 

Refined wheat flour of 650 type acquired from a local company, S.C. Mopan S.A. (Suceava, Romania), was used in this study as control and was analyzed through the same analytical procedures previously described for the following characteristics: moisture (14.00%), protein (12.60%), fat (1.40%), ash (0.65%). The wet gluten content (30.00%) of wheat flour was determined manually according to methods stated in the ICC method 106/1, falling number index (312 s) through ICC 107/1, and gluten deformation index (6.00 mm) by the Romanian standard method SR 90:2007, respectively. Sodium chloride (S.C.SANOVITA S.R.L., Vâlcea, Romania) and dry yeast (S.C. ROMPAK S.R.L., Pașcani, România) were also used in the breadmaking.

### 2.2. Amaranth Flour Fractionation

Amaranth seeds were ground with laboratory machinery (Grain Mill, KitchenAid, Model 5KGM, Whirlpool Corporation, Benton Harbor, MI, USA ) and sieved with a Retsch Vibratory Sieve Shaker AS 200 basic (Haan, Germany) for 30 min, at 70 Hz amplitude, in order to obtain the amaranth flour particle size > 500 μm. The amaranth flour (AF) particle sizes were coded as integral amaranth flour, AI (before sieving), large, AL (>300 μm, >500 μm), medium, AM (>180 μm, <300 μm) and small, AS (<180 μm), and stored in air-tight zip-lock pouches for further analysis.

### 2.3. Proximate Composition and Color Parameters of Amaranth Flours

Amaranth flours (AI, AL, AM, and AS) were analyzed according to the ICC methods as previously described: moisture content (110/1), protein content (105/2), fat content (104/1), and ash content (105/1), while total carbohydrate content was determined by difference [13].

The color parameters of integral amaranth flour (AI) and each particle size (AL, AM, and AS) in terms of lightness (*L**), the intensity of green (negative) or red (positive) (*a**), and intensity of blue (negative) or yellow (positive) (*b**) were determined with a CR-700 colorimeter (Konica Minolta, Tokyo, Japan). Each parameter was measured in triplicate.

### 2.4. Functional Properties of Amaranth Flours

Water absorption capacity (WAC), water retention capacity (WRC), and swelling capacity (SC) of the amaranth flour particle sizes were determined by procedures reported previously by Coțovanu and Mironeasa [13]. The water absorption capacity was determined using one g amaranth flour at each particle size which was mixed with 10 mL of distilled water, kept at ambient temperature for 30 min, and centrifuged for 10 min at 2000 rpm. WAC was reported as g of water retained per one gram of sample. Water retention capacity was measured using two grams of each amaranth flour, which was weighed into test tubes, and 20 mL of distilled water. Samples were vortexes and allowed to stand for 1 h at 23 °C before centrifuging at 1600 rpm for 25 min. Water excess (supernatant) was decanted and samples were allowed to drain. The weight ratio of dissolved solids in the supernatant and dry samples reports the WRC. Swelling capacity was determined using one gram of each amaranth flour which was mixed with 10 mL of water in a centrifuge tube. This weighed tube was heated in a water bath at 80 °C for 15 min. After this operation, the tube was centrifuged at 2000 rpm for 30 min and the supernatant was decanted and then weighed. SC is the percentage of swelling per gram of flour. Bulk density (BD) of AF particle sizes was determined according to the method described by Ikegwu [18]. The sample (50 g) was placed in a 50 mL graduated cylinder and it was operated until no differences were observed. The BD was determined as the weight of flour (g) per flour volume (mL).

### 2.5. Flour Samples’ Formulation

Wheat flour was mixed with amaranth flour at different addition levels (5%, 10%, 15%, and 20%) and particle sizes (AL, AM, and AS) for 30 min in a Yucebas Y21 machine (Izmir, Turkey) in order to obtain the wheat–amaranth composite flour (AL_5, AL_10, AL_15, AL_20, AM_5, AM_10, AM_15, AM_20, AS_5, AS_10, AS_15, and AS_20). The sample without amaranth flour addition was considered as the control.

### 2.6. Proximate Composition and Color Parameters of Wheat-Amaranth Composite Flours

The composite flours formulated were analyzed according to the ICC methods as previously described: moisture content (110/1), protein content (105/2), fat content (104/1), and ash content (105/1), while total carbohydrate content was determined by difference [13].

The color parameters of composite flours, *L*, a*,* and *b** were determined with a CR-700 colorimeter (Konica Minolta, Tokyo, Japan). Each parameter was measured in triplicate.

### 2.7. Dough and Bread Making Process

Composite flour (300 g), salt (1.8 g/100 g), and *Saccharomyces cerevisiae* type yeast (3 g/100 g) were used in the bread-making recipe. The water amount was calculated based on the water absorption capacity of the flour tested on the Mixolab (Chopin, Tripetteet Renaud, Paris, France). The bread dough was prepared using an indirect method. A half amount of composite flour and total amount of yeast and water were mixed to form the sourdough which was fermented at 30 ± 2 °C and 85% relative humidity (RH) for 2 h in a fermenting chamber (PL2008, Piron, Cadoneghe, Padova, Italy). After this time, fermented sourdough and the remaining portion of the composite flour with salt were kneaded in a professional mixer for 10 min (Kitchen Aid, Whirlpool Corporation, Benton Harbor, MI, USA) and then the dough was fermented at 30 ± 2 °C and 85% relative humidity (RH) for 1 h in the same fermenting chamber. When the dough fermentation was complete, each dough sample was divided into 400 g, molded, and transferred into aluminum trays (size about 10 × 10 × 10 cm) for 1 h (30 ± 2 °C and 85% RH) in the fermenting chamber for the final fermentation of sugars. Then, the bread was baked at 220 ± 5 °C for 25 min in an oven (Caboto PF8004D, Cadoneghe, Padova, Italy). After baking, the bread was subjected to analysis after cooling for 2 h.

### 2.8. Dough Rheology and Texture Profile Analysis

The rheological behavior of composite flour dough was analyzed using a Chopin Alveograph NG-97 (Tripette et Renaud, Villeneuve-la-Garenne, France) with the standard procedure AACC 54-30 (AACC, 2000). Under this test, the dough is stretched during the biaxial extension stage until rupture and the recorded normal force gives information about dough firmness and extensibility. The main alveographic parameters registered were: dough tenacity, P (mm H_2_O); dough extensibility, L (mm); index of swelling, G; dough strength, W (10^−4^ J); and curve configuration ratio, P/L.

For the texture profile analysis (TPA), the samples were prepared as for breadmaking, and 50 g of the fermented dough was weighed and tested with a TVT-6700 texture analyzer (Perten Instruments, Hägersten, Sweden). A 35 mm diameter cylindrical probe was used in a double-compression test to compress the samples up to 50% of the original height at a test speed of 5.0 mm/s, trigger force of 20 g, and the interval time between two compressions was 12 s, according to the method used by Mironeasa et al. [19]. Hardness (the maximum force registered in the first compression cycle), adhesiveness (the negative area obtained in the first cycle), springiness (the distance of the detected height of the second compression), and cohesiveness (the ratio between the positive area of the second cycle and the positive area of the first cycle) [19] were registered from the TPA curves. The measurements were taken three times for each sample.

### 2.9. Evaluation of Bread Physical Characteristics

The bread samples were analyzed according to the Romanian standard procedure SR 91:2007, and the following parameters were determined: loaf and specific volume, porosity, elasticity, crumb, and crust color. The specific volume was measured by the rapeseed displacement method (cm^3^/g) and it was calculated based on the loaf volume (cm^3^) and mass (g). Porosity (%) was determined by using Equation (1). A 45.50 diameter and 60 mm height cylinder obtained from the middle of the bread was weighed, and its volume was measured. For the bread elasticity measurements, the cylinder was weighed, pressed for 1 min until half of the cylinder height was achieved, then the force was removed and the bread sample was left to recover for 1 min, and the final height was measured. Elasticity (%) was given by the difference between the initial and final height [20]:(1)$$Porosity\;\left(\%\right)=\frac{V-\frac{m}{\rho }}{V}\;\cdot 100$$
where *V* is the crumb cylinder volume (cm^3^), m is the crumb cylinder weight in (g), and ρ is the crumb density in (g/cm^3^).

Crust and crumb color parameters (*L**, *a**, and *b**) were also determined using a CR-700 colorimeter (Konica Minolta, Tokyo, Japan). All analyses were performed three times.

### 2.10. Bread Crumb Texture

The texture profile analysis (TPA) of bread was determined with a TVT-6700 texture analyzer (Perten Instruments, Hägersten, Sweden), in accordance with the method described by Iuga et al. [21]. A 25 mm diameter cylindrical probe was used in a double-compression test up to 20% of the original sample height at a test speed of 1.0 mm/s, trigger force of 5 g, and the interval time between two compressions was 15 s. Firmness, springiness, gumminess, and cohesiveness were registered from the curves’ TPA. For each sample, three repetitions were performed.

### 2.11. Statistical Analysis

Software SPSS 25.0 (trial version) (IBM, New York, NY, USA) was used for statistical analysis of the data. Statistically significant differences for functional properties of amaranth flour particle sizes were determined by one-way ANOVA. A two-way ANOVA with Tukey’s test at 95% confidence level was also applied to test significant differences for the other parameters. A principal component analysis (PCA) was applied to observe the relationships between the wheat–amaranth flour proximate composition, dough and bread characteristics, and to visualize similarities or dissimilarities between them.

## 3. Results

### 3.1. Chemical Composition, Functional Properties, and Color Parameters of Amaranth Flours

The values of the chemical composition of the large (AL), medium (AM), and small (AS) amaranth particle sizes compared with the integral amaranth flour (AI) are shown in Table 1. The moisture content of the amaranth flours presented significant (*p* < 0.05) differences between particle sizes, which decreased with the reduction of the particle size. It can be observed that there are no differences regarding the moisture content of integral amaranth flour (AI) and large particle size (AL). The higher shear force that occurs during grinding affects the small and medium particle sizes especially, which can explain the variation in moisture content. The protein content of amaranth flour ranged between 9.22–29.35% and presented significant differences (*p* < 0.05) between all the samples. The highest protein content was found in the amaranth small particle size (29.35%), followed by the medium particle size (25.33%). Comparing the integral amaranth flour and the large particle size, there are significant differences, a fact that can be explained by the localization of protein in the germ of the seed (65%) [4,5]. The high level of protein content from medium and small fractions indicates that these particle sizes could be useful for developing new protein-enriched products, with different amaranth varieties being a complementary protein source [22].

The fat content of the amaranth flours, integral, and particle sizes varied between 7.11–8.09% and were significantly (*p* < 0.05) different between all samples. The results are in line with those previously obtained by some authors [23] on whole amaranth flour, and also on some different amaranth particle sizes obtained by dry milling [24,25]. A fine grinding will disintegrate rigid hull that protects the amaranth seed, favoring lipid extraction [26,27], which is a rich source of linoleic (C18:2, ω-6) and linolenic acids (C18:3, ω-3) [28].

Regarding the ash content of amaranth flours, which ranged between 1.05–4.45%, the same significant difference (*p* < 0.05) was observed between samples as in the case of the protein content. The small and medium particle sizes presented the highest contents of ash, probably due to the high content of minerals, the ash content being closely related to the morphological structure of amaranth seed. High levels of K, P, Ca, Mg, Na, and Fe can provide health benefits for preventing osteopenia and osteoporosis disease [29,30]. In the integral amaranth flour, a low content of ash was found compared with the amaranth flour particle sizes. The values found are in the range of those reported by other authors [24].

Comparing amaranth with small particle sizes of quinoa and buckwheat obtained in the same conditions, a higher quantity of proteins, fats, and ash was observed in amaranth small particle sizes [12,13].

The carbohydrate content of the studied amaranth flours varied greatly from 49.73 to 71.13% and presented significant (*p* < 0.05) differences between samples. It was observed that AI had a higher carbohydrate content (71.13%), the result being in line with other authors’ reports [23], while with the reduction of particle size, the carbohydrate content decreased, a trend that was also previously observed [24].

The results for the functional properties of amaranth flours are presented in Table 1. Water absorption capacity (WAC) is an important parameter of food products that affect economic profitability and quality. WAC represents the power of protein and fiber to absorb and retain water providing hydrophilic parts such as polar and charged side chains [13]. The WAC values of amaranth flour particle sizes varied between 2.37% and 3.08% and showed a significant difference (*p* < 0.05) between samples. However, particle AS presented the highest WAC, which may be directly correlated with higher protein and ash content of the smaller particle size and indirectly correlated with the fat content of this particle, while AI had the smallest WAC and the lowest content of protein and ash. Data results were in agreement with those reported by Olawoye and Gbadamosi [31] and Shevkani et al. [32] for amaranth flours. Regarding water retention capacity (WRC), amaranth flour particle sizes only showed significant differences (*p* < 0.05) between the small particle (AS) and the other particle sizes (AM and AL) and AI. The AS presented the highest WRC (5.33%) which can probably be explained by amaranth seed morphology, which presented high ash and protein content and a low content of fats. It is known that the ability of flour to associate with water, at room temperature, depends mainly on proteins [33]. Swelling capacity (SC), defined as the ability of the starch granule to swell freely during heating in water, showed that the amaranth flour particle sizes presented values between 4.18 and 6.60 mL/g. The SC showed significant differences (*p* < 0.05) among all particle sizes and integral amaranth flour (Table 1), with the highest value being found in the small particle size, a fact that can be explained due to the increased level of damaged starch from this particle. The hydration properties of amaranth flour decreased when the particle size decreased, which can be partially explained by the larger surface area and volume pore of these particles. These results are consistent with those obtained by Alarcón-García et al. [34]. Bulk density (BD) of amaranth flour particle size values varied between 0.35 and 0.76 (g/mL) and all amaranth flours presented significant differences (*p* < 0.05) between them. The hydration capacities of wheat flour, regarding the value of water absorption capacity, water retention capacity, and swelling capacity, presented the following values: 1.72 (%), 4.74 (g/g), and 4.90 (mL/g), while bulk density was 0.52 (g/mL).

The measure of food color is a quality parameter when new ingredients are added. Color parameters of integral amaranth flour (AI), large (AL), medium (AM), and small (AS) particle sizes are presented in Table 1. Lightness (*L*)* did not show significant differences between integral amaranth flour and the large and medium particle sizes, whereas the small particle size was brighter. The *a** values presented statistical (*p* < 0.05) differences only between AI and the large particle size (AL), and the medium and small one (AM, AS), while *b** values did not present significant (*p* < 0.05) difference between the samples. This non-significant effect could be explained by the cream color of the amaranth flour which shows similarity with that of the integral wheat flour. The color parameters of basic materials may not always be the only ones responsible for the final food color. During the thermal process, the food products are exposed to biochemical reactions that may alter color properties, especially Maillard reactions and the caramelization process.

### 3.2. Physicochemical Properties of Wheat-Amaranth Composite Flours

The impact of different addition levels and particle size of AF on the physicochemical value of the wheat-amaranth composite flours used in this research is shown in Table 2. The results of two-way ANOVA analysis indicated that both factors (level of AF addition and particle size) and their interaction have a significant (*p* < 0.001) effect on all the physicochemical properties and color parameters of the formulated samples.

Composite flours presented significantly (*p* < 0.001) lower moisture contents than the control sample (Table 2). The moisture values decreased when AF addition increased and particle size decreased in the wheat–amaranth composite flour. This fact may be due to the grinding process of the AF without prior conditioning, with higher shear force, creating a higher temperature that leads to moisture loss.

The small and medium particles of AF added to wheat flour led to an increase in protein content with the addition level compared with the large particles, a trend that can be explained by the low content of proteins in large particles. For the samples with large particle size, protein content decreased with the addition level. The results are related to the amaranth seed protein which is different from other cereals because 65% it is located in the germ and 35% in the endosperm, compared with other cereals, in which it is located 15% in the germ and 85% in the endosperm [35].

The fat content of wheat-amaranth composite flours was significantly (*p* < 0.001) influenced by the amount and particle size of AF added and showed an increase with the addition level compared with the control (Table 2). The highest content of lipids was observed for the composite flour with the medium particle size, followed by the sample with large particles, while the composite flour with small particles presented the lowest values. The increase in fat content with the increase in the AF level can be explained by the complex compounds formed between fat, cell wall materials, and proteins, where the most localization of medium particle size is in the embryo fraction. These data are in accordance with previous studies reported by Alonso-Miravalles and O’Mahony [36] and Alvarez-Jubete et al. [37] which reported that amaranth fats are two to three times higher than in other cereals. The fact that the wheat-amaranth composite flour is rich in lipids, improves the technological process when these protein-rich flours are used as ingredients (e.g., baked products) because lipids are important for texture and flavor [27].

Ash content significantly (*p* < 0.001) increased in every sample, and was observed to be a rising trend for both factors—addition level and particle size—compared with the control. This increase in ash content with an increase in the AF amount and particle size can be related to the high content of ash from amaranth whole grain because in whole grains bran contains higher mineral contents than endosperm [38]. The higher contribution of bran to the wheat-amaranth composite flour could be explained by the smaller size of amaranth grain [39], which gives a larger surface area of bran per unit amount of grain, compared with whole wheat [40].

Carbohydrates from composite flours are significantly (*p* < 0.001) different and decreased when supplementation of wheat flour with AF increased and particle size decreased. The addition of AF to the wheat flour significantly (*p* < 0.001) affected the carbohydrates content (Table 2) and this fact can be explained by the chemical composition of amaranth grain, where carbohydrates are found usually in lower amounts than in cereals [37].

The color parameters (*L**, *a**, and *b** values) of the composite flours (*p* < 0.05) varied significantly with the addition level and particle size of AF added to the wheat flour (Table 1). Particle sizes showed various effects on composite flour and resulted in a decrease in lightness (*L**) for all samples when the addition level increased and the samples were therefore characterized by a lighter color. The *a** values presented a statistically (*p* < 0.001) difference between samples, and it can be observed in Table 2, that *a** values increased when AF level and particle size increased. The *b** value did not show a similar trend, but in general, the values increased with the amount of AF added and decreased with particle size reduction. The yellow color could be attributed to the presence of carotenoid pigments. These results are in line with those found by Sanz-Penella et al. [8].

### 3.3. Alveographic Parameters of Composite Flour Dough

Dough alveographic results are shown in Table 3, and all factors, level of AF addition, type of particle size, and their interaction had statistically significant (*p* < 0.001) effects on dough rheological parameters.

Replacing the gluten network from wheat flour with amaranth flour to develop improved bakery products is a challenge for grain technologists. The alveograph test simulated dough transformation during processing (sheeting, rounding, and molding) providing useful information for processors. For breadmaking operations, it is important to establish the extensibility of dough, which represents the most unique property of wheat dough for obtaining the characteristic structure and volume of the baked products [41]. High extensibility is necessary to obtain bread with high volumes, which is the desired attribute, so that the final product will have high crumb porosity with a soft crumb texture [42].

For dough tenacity (P), all factors, level of AF addition, type of particle size, and their interaction are statistically (*p* < 0.001) significant (Table 3). P was found to increase significantly with the increase in AF addition. The rise of AF increased dough tenacity (P) in all composite flour dough samples, compared with the control dough, except the AS_5 dough, where P was smaller than the control dough sample. For these parameter values, there was a difference between the AF particle sizes when they were added in different amounts in wheat flour dough. This increase in dough tenacity may indicate that non-gluten flour incorporated into wheat flour significantly increased dough tenacity, as was previously demonstrated by some authors [43,44].

Table 3 shows that the addition level, particle size, and the interaction between them, significantly affected (*p* < 0.001) the dough extensibility index (L), and a decrease in dough extensibility (L) was observed when the AF amount and particle size increased, compared with the control dough, which may be explained by the gluten dilution effect. The findings that L decreased as the AF addition level and particle size increased are in accordance with data from the literature [44] and with other studies, which incorporated AF in the dough, noticing a decrease in the elasticity of dough [45,46]. In the control sample dough and samples with lower AF addition level, dough extensibility was higher. L increased significantly (*p* < 0.001) with the reduction of particle size. A small particle size in dough absorbs water easier, due to the larger specific surface area of amaranth starch [11], and facilitates the integration of the small particles into the dough network. The index of swelling (G) recorded dough extensibility obtained from the square root of the volume of air, in millimeters, required to inflate a dough bubble until it ruptures [38]. There was a significant (*p* < 0.001) difference for AF addition level, type of particle size, and interaction between them. Table 3 shows that G decreased, in the same way as L, when AF content and particle size increased, compared with the control dough. This proves that dough extensibility decreased when the AF amount increased, due to the reduced water availability and weakening of the gluten network. The dough strength (W) provides important information about the quality of bakery products, and the optimum values vary depending on the type of product. Dough strength is highly influenced by the polymeric protein quantity and its size distribution [47]. Table 3 presented statistically (*p* ≤ 0.001) significant results for AF addition level and type of particle size on W of wheat-amaranth composite flour dough. An increase in AF in the composite flour is linked to a decrease in dough strength. A decrease in W when gluten content of the composite flours decreased was also observed and reported by other authors [48].

As shown in Table 3, the AF addition level, particle size, the addition level, and the type of PS interaction had a statistically significant (*p* < 0.001) effect on the P/L. The P/L ratio evaluated the concordance between dough tenacity and extensibility and played a key role in the technological process of leavened products [49]. The results of P and L parameters on P are in direct correlation with the P/L ratio value, which is conducted to identify unbalanced doughs. The increase in the AF amount in wheat flour dough significantly increased (*p* < 0.001) the P/L ratio, while the particle size increased the P/L ratio in an irregular order.

### 3.4. Dough Texture Profile Analysis

Texture analysis of dough is very important in the development and production of bread products. The intermediate product dough quality can be an indicator of the final product sensorial characteristics. The rheology and machinability of dough enrichments with non-gluten flours will be affected and will influence the final baked product. Figure 1 shows the effect of different addition levels and particle sizes on the textural properties of the composite dough.

The hardness, which is a positive character in products with low moisture content, showed higher values in dough containing AF compared with the control. Adhesiveness, which is the force required to remove food from any surface with which it has contact with, presented lower values compared with the control [49]. All particle sizes (AL, AM, and AS) revealed for dough hardness had a regular trend when levels of AF increased. The samples with 15% and 20% addition levels increased dough hardness compared with those with 5% and 10% replacements, which indicated an increase in dough strength. This behavior may be due to the chemical composition of AF that presents globular proteins of 11S and P-globulin type [50]. The specific proteins and fibers of these pseudocereals retain more water than gluten proteins and produce the dough gluten matrix due to the gelation process [50]. When the addition level of AF increased, gluten proteins were more diluted, meaning that the 11S and P-globulin from amaranth were not sufficient to form a gluten network in the mix, which resulted in a softer dough. Guardianelli et al. [50] reported similar results for wheat-amaranth flours dough. A decrease in adhesiveness when the addition level of AF increased suggested a strong interaction between the chemical composition of this flour (proteins, starch, and fiber) and water, leading to a decrease in dough hardness and consistency. Springiness describes the time a product’s crumb springs back to its undeformed state after a compressing force has been removed and is lost during storage, as the staling process advances [14]. This parameter presented a regular decrease compared with the control, but dough samples AL_10 and AM_15 had higher values than the control dough. Dough cohesiveness decreased when AF addition level increased and AF particle size decreased compared with the control, probably because lipids provided flour cohesion [51]. The direct correlation of dough cohesiveness with flour particle size could be due to the surface area increase and the formation of inter particulate bonding which provides more cohesive and less free-flowing powders [52]. The gliadins from wheat are mechanically characterized by little or no resistance to extension, therefore appearing to be responsible for dough’s cohesiveness, while the glutenins are responsible for the dough’s resistance to extension [53].

### 3.5. Technological Properties and Color Analysis of Composite Flour Bread

Due to the absence of gluten from amaranth flour, the loaf and specific volume decreased from 382.13 to 267.02 cm^3^ and from 2.53 to 1.83 cm^3^/g, respectively, as the amaranth flour amount increased (Table 4).

AF particle size, addition levels, and their interaction caused significant changes (*p* < 0.001) on bread technological parameters. According to these results, bread with higher amounts of all AF particle sizes presented lower specific volume, compared with the control. This negative effect could be due to the poor baking quality of the flour. The globular proteins (11S and 9P) diluted the gluten network and decreased the alfa amylase activity which decreased maltose availability for yeast in the proofing process. A higher bread volume was observed for bread samples with large and medium particle sizes, compared with the bread with small particle sizes. Bread with medium particles at addition levels of 5% and 10%, and also bread with particles large at 5% (AL_5), showed higher values for specific volume than the control bread. These results are according to those obtained by Ayo [54] for bread supplemented with amaranth flour. The porosity of bread was affected significantly (*p* < 0.001) by the addition of AF, particle size, and their interaction. The changes in crumb porosity may be related to its springiness and partly impacted the rating of crust and crumb appearance. When the AF addition level increased, the porosity of bread decreased significantly, while the AF particle size impacted the decrease in bread porosity in an irregular trend, as follows: AM, AL, and AS. The addition of AF, type of particle size, and the interaction between them, significantly affected (*p* < 0.001) bread elasticity. A regular trend was observed in the addition level of AF on bread elasticity, which decreased when the amount of AF increased (Table 4). Regarding the effect of AF particle size on bread elasticity, it was observed that the decrease in particle size led to a poor decrease in elasticity. Furthermore, it can be seen that all bread samples, except the samples with 20% addition, showed higher elasticity values than the control bread. This phenomenon can be related to the fact that amaranth flours are rich in dietary fiber and proteins (albumin), which interact with wheat glutenin protein through disulfide bonds, which do not weaken the gluten network overmuch [48]. On the other hand, the effect on volume and crumb texture could be attributed to the complexes formed between the high polar lipid content and starch, which contributes to stabilizing gas cells and is possible to improve bread elasticity [40,55]. All bread technological parameters decreased significantly (*p* < 0.001) compared with the control, and could be explained due to the weakening of the gluten network which leads to a decrease in volume, porosity, and elasticity in the formulations with 20%.

The results obtained regarding the technological properties of the bread are following those previously obtained by Sanz-Penella et al. [8] and Miranda-Ramos et al. [3].

The color parameters of the crust and crumb were presented in Table 5. The crust color of the bread after baking was significantly (*p* < 0.001) darker compared with the control bread.

Increasing the amaranth flour in the composite flours bread and decreasing AF particle size resulted in a significant (*p* < 0.001) decrease in the crust lightness (*L**). This was due to the high protein content in AF, which brought about the Maillard browning during baking. This effect may also be due to the darker color of the AF particle size, and especially the medium and large fractions compared with wheat flour. The crust *a** values ranged between positive values (red hue), increasing significantly (*p* < 0.001) when the proportion of AF increased and particle size decreased. The yellowness value (*b**) of the crust was significantly increased (*p* < 0.001) when the level of AF increased, and PS decreased, compared with the control, especially in formulations substituted with 15% and 20% of amaranth flour. These results are similar to the previous results of Rosell et al. [56] who found that the bread containing 12.5%, 25%, and 50% AF had a yellow-reddish crispy crust. Similarly, Sanz-Penella et al. [8] found that the darkness and redness of bread were increased when the amount of AF increased in the composite flour. Regarding crumb and color parameters, they were significantly (*p* < 0.001) affected by the AF addition level, the type of particle size, and the interaction between them. Bread crumb samples showed a tendency to decrease in terms of lightness (*L**) when the AF addition level increased, and also when the AF particle size decreased. These results indicated darker crumbs in composite flours bread compared with the control. This phenomenon can be explained by the direct correlation between the chemical composition of AF, due to the increase in protein content with lysine residues and reduced sugar content [57] that react during baking producing a non-enzymatic Maillard browning reaction, and due to the applied conditions during cooking (temperature, relative humidity, and transfer modes of heat) [58]. Similar findings were reported by Rosell et al. [56] and Miranda-Ramos et al. [3] when amaranth flours were used in various addition levels. The *a** values ranged between negative values (green hue), increasing significantly (*p* < 0.001) when the proportion of AF increased and particle size decreased. The yellowness value (*b**) increased significantly (*p* < 0.001) when the level of AF increased, and particle size decreased, compared with the control, especially in formulations with 15% and 20% of amaranth flour.

### 3.6. Bread Texture Profile Analysis

The food product texture is an essential parameter in determining product quality which determines its shelf-life. Firmness is defined as the first compression force, while springiness is the ratio of the first and second peak of force necessary to compress the sample [59]. Textural parameters of the formulation bread, enriched with different levels and particle size of AF are shown in Figure 2.

The bread firmness had a rise regular trend when the level of AF increased, but bread samples AM_5 and AM_10 had a lower value of hardness than the control. The sample with large particles at 15% and 20% addition have remarkably higher bread firmness. The increase in firmness with AF addition is related to the higher content of gluten-free flour which diluted the gluten network [8,60]. On the other hand, it was observed that samples with medium particle sizes at low addition (5% and 10%) presented lower firmness value than the control, showing that albumin proteins from amaranth grain can behave like gluten in the dough and may explain the results obtained, which are in accordance with the results previously obtained by Oszvald et al. [47].

Springiness is the level to which a crumb can return to its initial form after compression [10]. The changes that occurred showed that when amaranth flour addition level increased, the springiness of the bread increased, which might be explained by the high polar content of lipids in amaranth, and might act as a gas stabilizing agent when baking bread, thus improving the springiness of the final product [37]. These results are according to the results obtained by Sanz-Penella et al. [8] and Kurek and Krzeminska [60].

Gumminess is the energy needed to chew food before swallowing and is calculated by hardness x cohesiveness x springiness [10]. Figure 2b shows that gumminess increased with the increase in the AF addition level, especially for the 15% and 20% additions. There was a difference between the particle size when AL, AM, and AS were added in wheat flour dough at different amounts. Gumminess showed a regular trend for these particles as follows: M, L, and S. These findings are consistent with previous research using amaranth flour [56].

Cohesiveness is defined as the strength of the internal bonds responsible for the bread structure [10]. Figure 2b presents the variation of bread cohesiveness depending on the two factors, the AF addition level, and the particle size. Cohesiveness decreased when particle size increased and rose when the addition level increased.

### 3.7. The Relationship between the Proximate Composition of Wheat-Amaranth Flour, Dough Rheology, and Bread Parameters

The correlation between composite flour proximate composition, dough alveographic and textural parameters, and bread parameters are shown in Figure 3.

The principal component analysis (PCA) was used to highlight the influence of AF addition level and particle size on wheat-amaranth composite flour, dough, and bread analyzed variables. The two principal components explained 76.57% of the total variance (PC1 = 60.96% and PC2 = 15.60%). The PC1 was associated with composite flour moisture, lipids, ash, carbohydrates, dough alveographic parameters (P, L, G, W, and P/L), dough textural parameters (hardness, adhesiveness, and cohesiveness), and bread technological properties (volume, porosity, elasticity), while PC2 was associated with composite flour protein and bread firmness and gumminess. A high opposition between protein and carbohydrates, dough adhesiveness, and dough hardness, P and L alveograph parameters, P/L ratio and bread volume, bread cohesiveness, and bread firmness was observed.

Regarding bread samples, Figure 4 shows a good relationship between the control sample and all bread with a 5% AF addition level. Clustering of the AS_15, AS_20, and AM_20 bread can indicate the similarity between chemical composition, and rheological and textural parameters, highlighting at the same time the opposition with AL_20. The position of AL_20 in the graph (Figure 4) showed a positive association with bread firmness (Figure 3), suggesting that large particle sizes have a great contribution to bread firmness. Along the PC1 axis, there was an opposition between the samples formulated with large particle size at the 10% and 15% addition levels of amaranth flour (AL_10 and AL_15) in wheat flour and those with medium particle sizes (AM_10 and AM_15). The position of the AM_10 in the space of the principal component (Figure 4) showed a close association with bread cohesiveness (Figure 3), whereas AL 10 underlined the correlation with the carbohydrate content of the composite flour. The position of bread with a 20% small particle size of amaranth flour (AS_20) in the space of the principal component is associated with the high protein content for the wheat-amaranth composite flour.

Multivariate correlation analysis provides information on the significant (*p* ≤ 0.05) relationship between wheat–amaranth composite flour, dough, and bread features. By applying Pearson’s correlation analysis for proximate composition, rheological, and technological parameters, a series of correlation coefficients were found (0.56 < r > 0.99).

Significant positive correlations were obtained between moisture and alveographic parameters L (r = 0.83), G (r = 0.86), and W (r = 0.76), but were negative with dough tenacity, P (r = −0.61). Significant correlations at *p* > 0.0001 were obtained for lipids with moisture (r = −0.65) and G (r = 0.86), protein with ash (r = 0.95) and carbohydrates (r = −0.98), and ash with carbohydrates (r = −0.97), which can be explained by the high nutritional components of amaranth flour and its addition in wheat flour directly increasing these values. Similar observations on relationships within the chemical composition of wheat flour and alveographic dough parameters were reported by Codină and Mironeasa [61] and Codină et al. [62].

A positive correlation at *p* < 0.05 was obtained between dough tenacity and dough hardness (r = 0.56), dough extensibility (L) and dough adhesiveness (r = 0.81), and dough cohesiveness (r = 0.69), but a negative correlation with dough hardness (r = −0.70) was observed. P/L ratio showed a strong positive correlation with dough hardness (r = 0.86; *p* < 0.0001), but a negative relationship, *p* < 0.05, with dough adhesiveness (r = −0.78), bread elasticity (r = −0.76), porosity (r = −0.62), and volume (r = −0.62). Bread volume was positively correlated with cohesiveness (r = 0.76), and bread porosity (r = 0.99) and elasticity (r = 0.81; *p* < 0.05), but a negative relation was observed with bread gumminess (r = −0.56; *p* < 0.05) and dough hardness (r = −0.82), while bread gumminess was strongly correlated with dough springiness (r = 0.99; *p* < 0.0001), hardness (r = 0.71), and negatively with dough adhesiveness (r = −0.82). Similar results between dough textural properties and bread volume and texture properties were found by Martínez-Anaya [63] and Mironeasa et al. [64].

## 4. Conclusions

Amaranth flour fractionated on large, medium, and small particle sizes showed variation in moisture, protein, lipids, ash, and carbohydrates content. The partial replacement of wheat flour with medium and small amaranth flour particle sizes increased the amounts of protein with the increase in the addition levels of amaranth, due to the contribution in chemical constituents of these particles. Lipids and ash increased in composite flour for all addition levels and particle sizes, whereas carbohydrate content followed a reverse trend. The lowest content of carbohydrates was found in composite flour with 20% small particle size amaranth flour. A decrease in lightness (L*) for all the composite flours was obtained when the addition level increased, the lowest value was found for the small particle size at 20% addition level. Small particle size presented the highest water absorption capacity (3.08%), water retention capacity (5.33%), and swelling capacity (6.60%). The dough tenacity increased significantly for all the particle sizes with the increase in amaranth flour addition level over 5%, while dough extensibility decreased. The increase in AF level in the composite flour led to a decrease in the index of swelling and dough strength depending on the particle size, whereas the P/L ratio increased following an irregular trend. Besides the nutritional enrichment in protein, lipids, and ash contents at any addition level, the wheat–amaranth composite flour up to 10% addition level provided the best properties in baking quality (volume, porosity, elasticity, firmness, and springiness) at any particle size. The medium particle sizes at an addition level of up to 10% produced bread with a lower hardness than the control.

The separation of different particle sizes of amaranth flour led to different fractions rich in nutritional substances which can be used separately in the development of added-value food products. The evaluation of the combined effect of amaranth flour particle size and addition level on the composition of wheat flour, dough rheology, and bread quality offer valuable information on the processes for the development of new enriched nutritional products.

## Figures and Tables

**Figure 1 foods-10-01539-f001:**
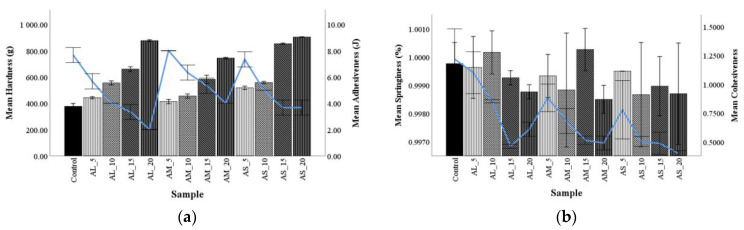
The effect of amaranth flour of large (AL), medium (AM), and small (AS) particle sizes added at 5%, 10%, 15%, and 20% on dough texture parameters: (**a**) hardness and adhesiveness and (**b**) springiness and cohesiveness compared with the control.

**Figure 2 foods-10-01539-f002:**
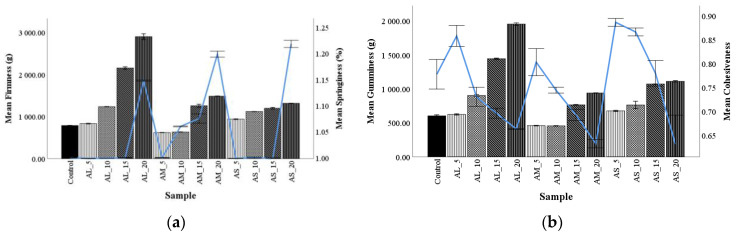
The effect of amaranth flour of large (AL), medium (AM), and small (AS) particle sizes added at 5%, 10%, 15%, and 20% on bread texture parameters: (**a**) firmness and springiness and (**b**) gumminess and cohesiveness compared with the control.

**Figure 3 foods-10-01539-f003:**
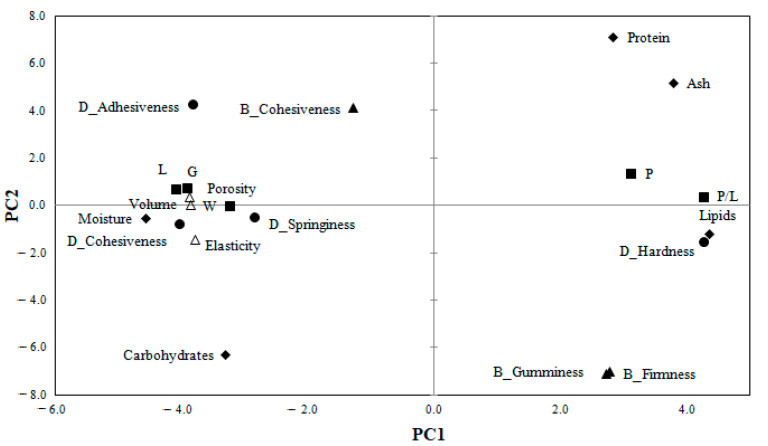
Principal component correlation loadings between wheat-amaranth composite flour chemical composition, dough rheological and textural parameters, and bread technological and textural parameters.

**Figure 4 foods-10-01539-f004:**
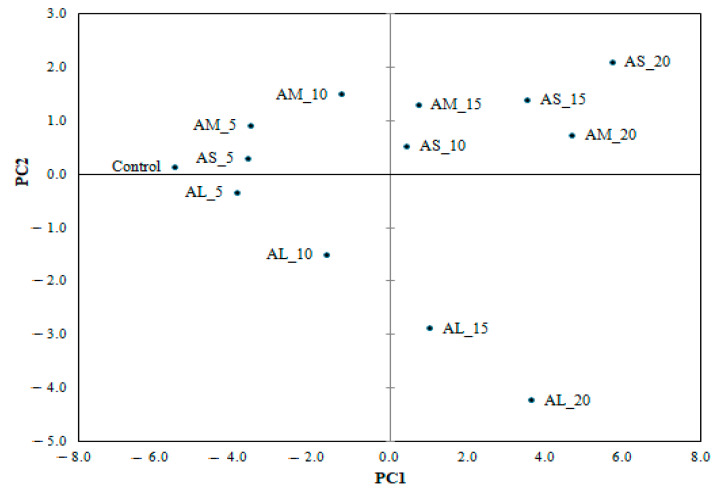
Principal component analysis of bread in function of the analyzed parameters as influenced by amaranth flour of large (AL), medium (AM), and small (AS) particle sizes added at 5%, 10%, 15%, and 20% compared with the control.

**Table 1 foods-10-01539-t001:** Chemical composition, functional properties, and color parameters of large (AL), medium (AM), and small (AS) particle sizes compared with the integral amaranth flour.

Parameters	AI	Particle Size		
		AL	AM	AS
**Chemical Composition**				
Moisture (%)	10.63 ± 0.04 ^a^	10.60 ± 0.04 ^a^	10.19 ± 0.04 ^b^	9.35 ± 0.04 ^c^
Protein (%)	9.22 ± 0.04 ^d^	10.18 ± 0.43 ^c^	25.33 ± 0.18 ^b^	29.35 ± 0.00 ^a^
Lipids (%)	7.95 ± 0.02 ^b^	7.48 ± 0.02 ^c^	8.09 ± 0.03 ^a^	7.11 ± 0.01 ^d^
Ash (%)	1.05 ± 0.01 ^d^	1.61 ± 0.02 ^c^	3.53 ± 0.02 ^b^	4.45 ± 0.03 ^a^
Carbohydrates (%)	71.13 ± 0.10 ^a^	70.11 ± 0.53 ^b^	52.85 ± 0.31 ^c^	49.73 ± 0.09 ^d^
**Functional Properties**				
WAC (%)	2.37 ± 0.05 ^c^	2.67 ± 0.03 ^b^	2.63 ± 0.14 ^b^	3.08 ± 0.09 ^a^
WRC (g/g)	3.68 ± 0.05 ^b^	3.78 ± 0.05 ^b^	3.89 ± 0.14 ^b^	5.33 ± 0.49 ^a^
SC (mL/g)	4.18 ± 0.10 ^d^	4.75 ± 0.01 ^c^	5.20 ± 0.01 ^b^	6.60 ± 0.02 ^a^
BD (g/mL)	0.76 ± 0.04 ^a^	0.58 ± 0.18 ^ab^	0.38 ± 0.01 ^bc^	0.36 ± 0.00 ^c^
**Color Parameters**				
*L**	77.43 ± 0.09 ^b^	77.72 ± 0.60 ^b^	77.01 ± 0.06 ^b^	79.13 ± 0.25 ^a^
*a**	−2.25 ± 0.10 ^a^	−2.32 ± 0.26 ^a^	−2.73 ± 0.14 ^b^	−2.98 ± 0.11 ^b^
*b**	26.07 ± 0.23 ^a^	26.50 ± 1.09 ^a^	25.93 ± 0.06 ^a^	26.03 ± 0.12 ^a^

Values are means ± SD. WAC—water absorption capacity; WRC—water retention capacity; SC—swelling capacity; BD—bulk density. *L**, *a**, and *b**—lightness, greenness, and yellowness, respectively. Values followed by the same superscript letter (^a^, ^b^, ^c^ and ^d^) at the same parameter are not significantly different according to Tukey’s HSD post hoc test (*p* ≤ 0.05).

**Table 2 foods-10-01539-t002:** Physicochemical properties and color parameters of wheat–amaranth composite flours compared with the control.

Sample	Moisture (%)	Protein (%)	Lipids (%)	Ash (%)	Carbohydrates (%)	Color Parameters		
						*L**	*a**	*b**
Control	14.08 ± 0.12 ^e^	12.45 ± 0.21 ^a^	1.41 ± 0.01 ^a^	0.69 ± 0.05 ^a^	71.36 ± 0.02 ^e^	91.46 ± 0.15 ^e^	−5.13 ± 0.04 ^a^	15.09 ± 0.07 ^a^
AL_5	13.83 ± 0.00 ^dz^	12.47 ± 0.01 ^bx^	1.70 ± 0.00 ^by^	0.70 ± 0.00 ^bx^	71.28 ± 0.02 ^dz^	91.16 ± 0.03 ^dy^	−5.27 ± 0.02 ^bx^	14.54 ± 0.01 ^bx^
AL_10	13.66 ± 0.00 ^cz^	12.36 ± 0.03 ^cx^	2.01 ± 0.00 ^cy^	0.75 ± 0.00 ^cx^	71.22 ± 0.03 ^cz^	90.54 ± 0.30 ^cy^	−4.91 ± 0.12 ^cx^	14.62 ± 0.11 ^cx^
AL_15	13.49 ± 0.00 ^bz^	12.23 ± 0.04 ^dx^	2.31 ± 0.00 ^dy^	0.79 ± 0.00 ^dx^	71.16 ± 0.05 ^bz^	89.99 ± 0.22 ^by^	−4.87 ± 0.07 ^dx^	15.49 ± 0.09 ^dx^
AL_20	13.32 ± 0.01 ^az^	12.11 ± 0.06 ^ex^	2.61 ± 0.00 ^ey^	0.84 ± 0.00 ^ex^	71.10 ± 0.07 ^az^	89.32 ± 0.66 ^ay^	−4.86 ± 0.00 ^ex^	15.53 ± 0.11 ^ex^
AM_5	13.81 ± 0.00 ^dy^	13.24 ± 0.01 ^by^	1.73 ± 0.00 ^bz^	0.79 ± 0.00 ^by^	70.42 ± 0.01 ^dy^	90.70 ± 0.04 ^dx^	−4.91 ± 0.04 ^by^	14.32 ± 0.07 ^by^
AM_10	13.62 ± 0.00 ^cy^	13.87 ± 0.02 ^cy^	2.07 ± 0.00 ^cz^	0.94 ± 0.00 ^cy^	69.50 ± 0.02 ^cy^	90.12 ± 0.19 ^cx^	−4.71 ± 0.15 ^cy^	14.83 ± 0.08 ^cy^
AM_15	13.42 ± 0.01 ^by^	14.51 ± 0.02 ^dy^	2.40 ± 0.00 ^dz^	1.08 ± 0.00 ^dy^	68.57 ± 0.03 ^by^	89.99 ± 0.22 ^bx^	−4.59 ± 0.07 ^dy^	15.29 ± 0.09 ^dy^
AM_20	13.24 ± 0.01 ^ay^	15.14 ± 0.03 ^ey^	2.73 ± 0.00 ^ez^	1.22 ± 0.00 ^ey^	67.65 ± 0.04 ^ay^	88.68 ± 0.52 ^ax^	−4.32 ± 0.08 ^ey^	15.34 ± 0.08 ^ey^
AS_5	13.77 ± 0.00 ^dx^	13.44 ± 0.00 ^bz^	1.69 ± 0.00 ^bx^	0.84 ± 0.00 ^bz^	70.27 ± 0.00 ^dx^	90.86 ± 0.26 ^dx^	−4.88 ± 0.09 ^by^	15.24 ± 0.04 ^by^
AS_10	13.54 ± 0.01 ^cx^	14.28 ± 0.00 ^cz^	1.97 ± 0.00 ^cx^	1.03 ± 0.00 ^cz^	69.19 ± 0.01 ^cx^	90.46 ± 0.12 ^cx^	−4.66 ± 0.02 ^cy^	15.15 ± 0.13 ^cy^
AS_15	13.30 ± 0.01 ^bx^	15.11 ± 0.02 ^dz^	2.26 ± 0.00 ^dx^	1.22 ± 0.00 ^dz^	68.11 ± 0.01 ^bx^	89.89 ± 0.19 ^bx^	−4.54 ± 0.00 ^dy^	15.09 ± 0.96 ^dy^
AS_20	13.07 ± 0.01 ^ax^	15.95 ± 0.01 ^ez^	2.54 ± 0.00 ^ex^	1.41 ± 0.00 ^ez^	67.02 ± 0.02 ^ax^	88.10 ± 0.13 ^ax^	−4.21 ± 0.02 ^ey^	15.09 ± 0.10 ^ey^
Two-way ANOVA *p* value								
Factor I: level of AF addition	*p* < 0.001	*p* < 0.001	*p* < 0.001	*p* < 0.001	*p* < 0.001	*p* < 0.001	*p* < 0.001	0.001
Factor II: type of PS	*p* < 0.001	*p* < 0.001	*p* < 0.001	*p* < 0.001	*p* < 0.001	*p* < 0.001	*p* < 0.001	0.004
Factor I x Factor II	*p* < 0.001	*p* < 0.001	*p* < 0.001	*p* < 0.001	*p* < 0.001	*p* < 0.001	*p* < 0.001	0.001

Mean values in the same column followed by different letters are significantly different (*p* < 0.05): a–e for AF different addition level containing samples (0–20%); x–z for AF different particle size containing samples (L, M, and S). *L**, *a** and *b** for lightness, greenness, and yellowness, respectively.

**Table 3 foods-10-01539-t003:** Alveograph rheological properties as affected by amaranth flour of large (AL), medium (AM), and small (AS) particle sizes added at 5%, 10%, 15%, and 20% compared with the control.

Type of Sample	P (mm H_2_O)	L (mm)	G	W (10^−4^ J)	P/L
Control	86.00 ± 0.50 ^b^	94.00 ± 3.00 ^e^	21.55 ± 0.35 ^e^	253.00 ± 4.00 ^d^	0.92 ± 0.35 ^a^
AL_5	88.00 ± 1.00 ^ax^	53.50 ± 0.50 ^dx^	16.30 ± 0.10 ^dx^	174.00 ± 2.00 ^cxy^	1.64 ± 0.03 ^bx^
AL_10	91.50 ± 0.50 ^cx^	47.00 ± 2.00 ^cx^	15.25 ± 0.35 ^cx^	162.50 ± 3.50 ^cxy^	1.95 ± 0.09 ^cx^
AL_15	94.00 ± 1.00 ^cx^	43.50 ± 0.50 ^bx^	14.70 ± 0.10 ^bx^	155.00 ± 0.00 ^bxy^	2.16 ± 0.05 ^dx^
AL_20	99.50 ± 0.50 ^dx^	31.50 ± 0.50 ^ax^	12.50 ± 0.10 ^ax^	130.5 ± 0.50 ^axy^	3.16 ± 0.07 ^ex^
AM_5	95.50 ± 0.50 ^ay^	61.00 ± 1.00 ^dx^	17.35 ± 0.15 ^dx^	198.50 ± 0.50 ^cy^	1.56 ± 0.03 ^bxy^
AM_10	104.00 ± 0.00 ^cy^	49.00 ± 1.00 ^cx^	15.55 ± 0.15 ^cx^	182.00 ± 2.00 ^cy^	2.12 ± 0.04 ^cxy^
AM_15	83.50 ± 0.50 ^cy^	43.50 ± 0.50 ^bx^	14.70 ± 0.10 ^bx^	129.50 ± 1.50 ^by^	1.93 ± 0.01 ^dxy^
AM_20	106.00 ± 1.00 ^dy^	28.00 ± 1.00 ^ax^	11.45 ± 0.55 ^ax^	117.50 ± 5.50 ^ay^	3.79 ± 0.17 ^exy^
AS_5	53.00 ± 0.00 ^ay^	97.50 ± 0.50 ^dy^	21.95 ± 0.05 ^dy^	147.50 ± 3.50 ^cx^	0.54 ± 0.00 ^by^
AS_10	95.50 ± 0.50 ^cy^	50.00 ± 0.50 ^cy^	15.80 ± 0.10 ^cy^	167.50 ± 3.50 ^cx^	1.89 ± 0.01 ^cy^
AS_15	111.00 ± 1.50 ^cy^	29.50 ± 0.50 ^by^	12.10 ± 0.10 ^by^	137.00 ± 4.00 ^bx^	3.78 ± 0.01 ^dy^
AS_20	116.00 ± 1.00 ^dy^	32.50 ± 1.50 ^ay^	12.70 ± 0.30 ^ay^	155.50 ± 3.50 ^ax^	3.57 ± 0.19 ^ey^
Two-way ANOVA *p* value					
Factor I	*p* < 0.001	*p* < 0.001	*p* < 0.001	*p* < 0.001	*p* < 0.001
Factor II	*p* < 0.001	*p* < 0.001	*p* < 0.001	0.001	*p* < 0.001
Factor I x Factor II	*p* < 0.001	*p* < 0.001	*p* < 0.001	*p* < 0.001	*p* < 0.001

Factor I—level of AF addition; Factor II—type of PS; Mean values in the same column followed by different letters are significantly different (*p* < 0.05): a–e for AF different addition levels containing samples (0–20%); x–z for AF different particle size containing samples (L, M, and S). P—dough tenacity; L—dough extensibility; G—index of swelling; W—dough strength; P/L—curve configuration ratio.

**Table 4 foods-10-01539-t004:** Technological properties of bread as influenced by amaranth flour of large (AL), medium (AM), and small (AS) particle sizes added at 5%, 10%, 15%, and 20% compared with the control.

Sample	Loaf Volume (cm^3^)	Specific Volume (g/cm^3^)	Porosity (%)	Elasticity (%)
Control	376.96 ± 0.98 ^e^	2.30 ± 0.06 ^c^	64.44 ± 0.31 ^b^	91.72 ± 0.07 ^b^
AL_5	382.13 ± 0.64 ^dy^	2.53 ± 0.05 ^dy^	68.83 ± 0.11 ^dx^	96.10 ± 0.95 ^dy^
AL_10	338.82 ± 1.51 ^cy^	2.27 ± 0.01 ^cy^	67.29 ± 0.00 ^cx^	95.55 ± 0.95 ^cy^
AL_15	326.45 ± 1.01 ^by^	2.14 ± 0.01 ^by^	66.01 ± 0.82 ^cx^	93.15 ± 0.51 ^by^
AL_20	301.20 ± 1.04 ^ay^	2.07 ± 0.02 ^ay^	52.24 ± 0.96 ^ax^	89.39 ± 0.33 ^ay^
AM_5	380.89 ± 0.51 ^dz^	2.47 ± 0.00 ^dyz^	69.74 ± 1.13 ^dz^	94.36 ± 0.90 ^dx^
AM_10	363.25 ± 0.56 ^cz^	2.36 ± 0.01 ^cyz^	68.73 ± 0.10 ^cz^	94.22 ± 0.94 ^cx^
AM_15	326.80 ± 1.58 ^bz^	2.14 ± 0.01 ^byz^	66.95 ± 0.12 ^cz^	91.33 ± 0.30 ^bx^
AM_20	307.96 ± 1.26 ^az^	2.11 ± 0.01 ^ayz^	64.43 ± 0.32 ^az^	86.99 ± 0.57 ^ax^
AS_5	354.68 ± 1.37 ^dx^	2.24 ± 0.03 ^dx^	66.56 ± 0.43 ^dxy^	96.27 ± 0.34 ^dx^
AS_10	340.45 ± 2.04 ^cx^	2.17 ± 0.01 ^cx^	64.30 ± 0.15 ^cxy^	93.05 ± 0.27 ^cx^
AS_15	310.94 ± 3.33 ^bx^	2.15 ± 0.01 ^bx^	65.60 ± 0.48 ^cxy^	90.10 ± 0.10 ^bx^
AS_20	267.02 ± 1.09 ^ax^	1.83 ± 0.05 ^ax^	58.86 ± 0.12 ^axy^	86.68 ± 0.02 ^ax^
Two-way ANOVA *p* value				
Factor I	*p* < 0.001	*p* < 0.001	*p* < 0.001	*p* < 0.001
Factor II	*p* < 0.001	*p* < 0.001	*p* < 0.001	*p* < 0.001
Factor I x Factor II	*p* < 0.001	*p* < 0.001	*p* < 0.001	*p* < 0.001

Factor I—level of AF addition; Factor II—type of PS. Mean values in the same column followed by different letters are significantly different (*p* < 0.05): a–e for AF different addition level containing samples (0–20%); and x–z for AF different particle size containing samples (L, M, and S).

**Table 5 foods-10-01539-t005:** Crumb and crust color of breads as influenced by amaranth flour of large (AL), medium (AM), and small (AS) particle sizes added at 5%, 10%, 15%, and 20% compared with the control.

Sample	Crumb Color			Crust Color		
	*L**	*a**	*b**	*L**	*a**	*b**
Control	72.3 ±0.27 ^d^	−4.48 ± 0.03 ^a^	19.02 ± 0.23 ^a^	67.69 ± 0.45 ^c^	0.78 ± 0.22 ^a^	31.60 ± 0.87 ^a^
AL_5	70.10 ± 0.49 ^bx^	−3.47 ± 0.16 ^bx^	20.48 ± 0.85 ^bx^	70.56 ± 1.09 ^cz^	1.81 ± 1.31 ^bx^	31.55 ± 2.09 ^bx^
AL_10	68.40 ± 0.83 ^cx^	−2.90 ± 0.31 ^bx^	20.56 ± 0.66 ^bx^	63.54 ± 0.24 ^bz^	1.69 ± 0.55 ^bx^	32.41 ± 0.57 ^cx^
AL_15	66.38 ± 0.53 ^abx^	−2.67 ± 0.15 ^cx^	20.99 ± 0.26 ^bcx^	63.22 ± 0.68 ^az^	1.90 ± 0.53 ^cx^	33.02 ± 0.57 ^dx^
AL_20	65.05 ± 0.33 ^ax^	−2.64 ± 0.05 ^cx^	23.82 ± 0.45 ^cx^	62.80 ± 0.65 ^az^	3.10 ± 0.34 ^dx^	35.62 ± 0.77 ^ex^
AM_5	68.03 ± 0.85 ^by^	−3.89 ± 0.18 ^bx^	19.37 ± 0.56 ^bx^	70.83 ± 0.64 ^cy^	6.27 ± 0.73 ^by^	31.59 ± 0.80 ^bz^
AM_10	66.37 ± 0.40 ^cy^	−3.26 ± 0.39 ^bx^	21.42 ± 0.59 ^bx^	62.67 ± 0.25 ^by^	6.31 ± 0.24 ^by^	35.84 ± 2.57 ^cz^
AM_15	66.14 ± 1.58 ^aby^	−2.87 ± 0.27 ^cx^	22.21 ± 1.21 ^bcx^	61.79 ± 1.88 ^ay^	6.71 ± 0.74 ^cy^	36.51 ± 0.73 ^dz^
AM_20	64.63 ± 0.59 ^ay^	−2.67 ± 0.16 ^cx^	23.82 ± 0.45 ^cx^	61.91 ± 1.35 ^ay^	6.90 ± 0.94 ^dy^	37.65 ± 1.02 ^ez^
AS_5	67.46 ± 0.52 ^bx^	−2.60 ± 0.27 ^by^	23.01 ± 0.37 ^by^	60.00 ± 4.50 ^cx^	6.76 ± 0.74 ^bz^	33.47 ± 0.99 ^by^
AS_10	66.19 ± 2.02 ^cx^	−2.58 ± 0.17 ^by^	23.60 ± 0.18 ^by^	59.39 ± 2.51 ^bx^	7.21 ± 2.05 ^bz^	34.59 ± 0.72 ^cy^
AS_15	65.02 ± 0.09 ^abx^	−1.93 ± 0.19 ^cy^	24.36 ± 0.51 ^bcy^	57.21 ± 2.27 ^ax^	7.45 ± 1.67 ^cz^	36.59 ± 1.08 ^dy^
AS_20	63.78 ± 0.86 ^ax^	−1.28 ± 0.12 ^cy^	25.00 ± 0.16 ^cy^	55.47 ± 1.90 ^ax^	8.36 ± 0.67 ^dz^	35.00 ± 0.74 ^ey^
Two-way ANOVA *p* value						
Factor I	*p* < 0.001	*p* < 0.001	*p* < 0.001	*p* < 0.001	*p* < 0.001	*p* < 0.001
Factor II	*p* < 0.001	*p* < 0.001	*p* < 0.001	*p* < 0.001	*p* < 0.001	*p* < 0.001
Factor I x Factor II	*p* < 0.001	*p* < 0.001	*p* < 0.001	*p* < 0.001	0.001	*p* < 0.001

Factor I—level of AF addition; Factor II—type of PS. Mean values in the same column followed by different letters are significantly different (*p* < 0.05): a–e for AF different addition level containing samples (0–20%); and x–z for AF different particle size containing samples (L, M, and S). *L**—lightness, *a**—greenness (negative) or redness (positive), and *b**—yellowness.

## Data Availability

Not applicable.

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
