# Peer review of "Impact of Different Amaranth Particle Sizes Addition Level on Wheat Flour Dough Rheology and Bread Features"

_foods, 2021, doi:10.3390/foods10071539_

Round 1

Reviewer 1 Report

The manuscript entitled: “Impact of different amaranth particle size addition level on wheat flour dough rheology and bread features” reports a study on the influence of amaranth at three different particle size in 650 type wheat flour for bread making. The topic is interesting nonetheless it is not new, and the novelty of the manuscript and contribution to te field of research shuld be better exploited. In first place, the use of different types of flours would have been useful to compare the results by adding amarant flour. The end points of the manuscript should be better assessed. There are a few remarks to address as detailed in the following. The experimental section should be more detailed. The “Basic materials” section reports data: did the Authors analyze the samples? Please give details with regard also to the nutritional added value due to the use of amaranth flour. The composition of the amarant flour and the wheat flour used should be given and analytical methods used detailed. If the wheat used was already “refined” please explain the need to further process s mentioned in the paragraph “Flour fractionation”. The paragraph 2.2 and following pnes should give details on the experimental procedures. The English language should be carefully checked also. The bread physical characteristics should be compared with international standard procedures (not only the Romanian ones). The composition data reported should be deteiled based on experimental determinations: please explain. The “nutritional enrichment”mentiond should be substantiated and discussed in greater detail. The novelty and contribution to the area of interest shoud be stressed and substantiated. A comparison with bread withouth amarant flous would be useful to better address the results and compare; more data are needed. Please, avoid to cite too dated References (e.g. see the one dated 1984, etc.) unless necessay and justified: use instead more recent references.

Author Response

The manuscript reports the results relating to the effects caused by the inclusion of increasing doses of amaranth flour with different particle sizes on the characteristics of the composite dough and bread. The work is aimed at the development of applications for new formulations wheat-amaranth bread recipe in the bakery sector.  There are some observations for the authors mentioned below.

  1. Add the functional characteristics of the wheat flour in respect to WAC, WRC, SC, BD, without including them in Table 1. Add a brief comment in 3.1

               We added briefly comment in 3.1. Section regarding the wheat flour functional properties values in respect to WAC, WRC, SC, and BD (Line 232-235). 

  1. Line 268: add a comment for b *

We added at line 268 a few sentences in order to explain the variance of b* parameter (Line 288 -290).

  1. Line 414-416. Check the caption of table 5, I found a typo.

The caption of Table 5 was corrected (Line 410-411).

  1. In general, the statistical analysis shown in the tables is not very easy to understand as regards the statistical difference in the columns between levels of additions and particle size.

We worked very carefully to obtain the statistical analysis from tables, using two-way ANOVA, testing significant differences for parameters studied. Regarding the first factor, AF addition level, mean values in the same column followed by different letters (a-e) are significantly different (p < 0.05) and the significant differences regarding the second factor, type of particle size  are followed by different letters (x-z). We would like to allow us the data obtained to be presented in this form.

               We would like to thank the referee for the close reading and for all the given comments suitable for improving the manuscript. The manuscript was modified according to the suggestions of the reviewer.

Reviewer 2 Report

The manuscript reports the results relating to the effects caused by the inclusion of increasing doses of amaranth flour with different particle sizes on the characteristics of the composite dough and bread. The work is aimed at the development of applications for new formulations wheat-amaranth bread recipe in the bakery sector.  There are some observations for the authors mentioned below.

Add the functional characteristics of the wheat flour in respect to WAC, WRC, SC, BD, without including them in Table 1. Add a brief comment in 3.1

Line 268: add a comment for b *

Line 414-416. Check the caption of table 5, I found a typo.

In general, the statistical analysis shown in the tables is not very easy to understand as regards the statistical difference in the columns between levels of additions and particle size.

Author Response

The manuscript entitled: “Impact of different amaranth particle size addition level on wheat flour dough rheology and bread features” reports a study on the influence of amaranth at three different particle size in 650 type wheat flour for bread making. The topic is interesting nonetheless it is not new, and the novelty of the manuscript and contribution to te field of research shuld be better exploited.

  1. In first place, the use of different types of flours would have been useful to compare the results by adding amarant flour.

We chose to study the addition of amaranth flour at different particle size, because this topic has not been studied before. The addition of wholemeal amaranth flour on different types of flours has been previously studied by other authors. Literature data show that different fractions of cereal grains have high biological value, depending on their anatomical structure, which is important to study and to include in bakery products. We added a few sentences in order to explain why amaranth flour particle size is useful for the baking industry and why processors could utilize certain particle sizes to obtain the desired product consistency in formulation with the appropriate rheological parameters and quality finite products (Line 521-524).

  1. The end points of the manuscript should be better assessed.

We really appreciate your suggestion in order to make a better manuscript from the scientific point of view. We reformulated the Conclusions section (Line 506-524). Also, we highlighted the strong correlation between the amaranth flour particle size added in wheat flour and the chemical composition of composite flour and with the technological properties of breads (Line 476-483).

  1. There are a few remarks to address as detailed in the following. The experimental section should be more detailed. The “Basic materials” section reports data: did the Authors analyze the samples?

Thank you for your referee. We detailed the experimental section. The basic materials was analyzed by us and we described more in detail the analytical procedure (Line 68-105).

  1. Please give details with regard also to the nutritional added value due to the use of amaranth flour.

In the introduction, we described from literature data the amaranth flour nutritional value and its importance on health-promoting values and benefits (Line 34-37). Also, we added in Table 1, new information about the chemical composition of integral amaranth flour (AI) and of each particle size (AL, AM, AS), and at 3.1. subsection we discuss the nutritional added value of each amaranth flour, regarding chemical composition (Line 178-214).

  1. The composition of the amarant flour and the wheat flour used should be given and analytical methods used detailed.

We detailed the analytical methods used for proximate composition of the amaranth seed and wheat flour (Line 68-78).

  1. If the wheat used was already “refined” please explain the need to further process s mentioned in the paragraph “Flour fractionation”. The paragraph 2.2 and following pnes should give details on the experimental procedures.

The refined wheat flour was aquired from a local company S.C. Mopan S.A. (Suceava, Romania). In the paragraph 2.2. “Flour fractionation” we described already the experimental procedures, but only the amaranth seeds were ground with a laboratory machinery and then sieved with a Retsch Vibratory Sieve Shaker, and we modified the subtitle 2.2 from “Flour fractionation” in “Amaranth flour fractionation” (Line 81). Also, we completed the experimental procedure (Line 83-84).

  1. The English language should be carefully checked also.

The English was revised by a native speaker.

  1. The bread physical characteristics should be compared with international standard procedures (not only the Romanian ones).

We used Romanian Standard because are aligned with international ones.

  1. The composition data reported should be deteiled based on experimental determinations: please explain.

We detailed the analytical methods used for proximate composition of the wheat-flour composite flour (Line 68-73).

  1. The “nutritional enrichment”mentiond should be substantiated and discussed in greater detail. The novelty and contribution to the area of interest shoud be stressed and substantiated.

We detailed in the introduction the importance of amaranth flour, the benefits it has on the health of the consumer, and we underlined in 3.1. the importance of using ground fractions of amaranth flour, compared to integral amranth flour, as well as their different biological value depending on the anatomical part of the amaranth grain (Line 177-207).

  1. A comparison with bread withouth amarant flous would be useful to better address the results and compare; more data are needed.

We compared all composite flours, doughs and breads with the wheat flour, dough and bread, which was used as Control sample.

  1. Please, avoid to cite too dated References (e.g. see the one dated 1984, etc.) unless necessay and justified: use instead more recent references.

We replaced the references (dated 1984) with a new one (Line 655-657).

Reviewer 3 Report

The manuscript reports a study to investigate the effects of two major factors: amaranth flour inclusion level (0, 5, 10, 15 and 20) and amaranth flour particle size (large, medium and small), and their interactions, on chemical composition of the composite flour mixture (basically bread composition) and physical properties of dough and bread.   Although the effect of amaranth inclusion level on bread making has been investigated previously by other investigators, the effect of particle size and its interaction with inclusion level is the main merit for the present study.  Therefore, the manuscript provides some new information.   The experiment was well designed and executed, data was statistically treated and presented well, and the manuscript is easy to follow.  

This reviewer recommends its publication after the authors address the following concerns.  

L106: 2.5. Proximate composition of wheat-amaranth composite flours

Color parameters are also included in this subtitle, which do not relate to proximate composition.  Therefore, the subtitle should be modified as “Proximate composition and color parameters of ….”

Table 1:  Since the table has sufficient spaces, please spell out all abbreviations

L199: “which can be explained, probably, by ist morphology”  what is “ist”?

L213: “3.2. Physico-chemical composition of wheat-amaranth composite flours” Change “composition” to “properties”.  There is no “physical composition” thermology.  This is also to be consistent with the wording in Line 218.   Alternatively, you can change “Physico-chemcial composition of…” to “Chemical composition and color parameters of….”

Table 2 should be proceeded with another table (a new table), showing chemical composition and color parameters of amaranth (before particle size reduction, serve as a control), AL, AM and AL samples.  This new table will help support discussion in Line 223-269, and many other places.  It is a pity that the authors overlooked this important aspect.  This missing piece of information is rather important for improving the quality of the manuscript. The rationale is that the particle size effect is magnified by two important properties of flour with difference particle sizes: chemical composition and physical properties (such as color, particle size itself, and water holding capacity, etc.)

L479:  Subtitle of 3.7, change to “wheat-amaranth proximate composition flour” to “proximate composition of wheat-amaranth composite flour”

L536-537:  “while higher level (15-20%) worsened dough rheology, which resulted also dependent upon flour particle size.”  This sentence is not clear. Need rewording.  Furthermore, authors need to add one or two sentences for summarizing how particle size affected dough and bread quality.  Since the main sale point of the article is adding particle size as another dimension (in addition to inclusion level), the conclusion should elaborate the effect of particle size as well as its interaction with inclusion level.

Author Response

The manuscript reports a study to investigate the effects of two major factors: amaranth flour inclusion level (0, 5, 10, 15 and 20) and amaranth flour particle size (large, medium and small), and their interactions, on chemical composition of the composite flour mixture (basically bread composition) and physical properties of dough and bread.   Although the effect of amaranth inclusion level on bread making has been investigated previously by other investigators, the effect of particle size and its interaction with inclusion level is the main merit for the present study.  Therefore, the manuscript provides some new information.   The experiment was well designed and executed, data was statistically treated and presented well, and the manuscript is easy to follow.

This reviewer recommends its publication after the authors address the following concerns.

  1. L106: 2.5. Proximate composition of wheat-amaranth composite flours

Color parameters are also included in this subtitle, which do not relate to proximate composition.  Therefore, the subtitle should be modified as “Proximate composition and color parameters of ….”

               Thank you for your kind remark. We replaced the subtitle as per your suggestion (Line 114).

  1. Table 1: Since the table has sufficient spaces, please spell aut

               We completed Table 1 with all data (chemical composition, functional properties, and color parameters) for integral amaranth flour (AI) and all three amaranth flour particle sizes (AL, AM, and AS) (Line 208-211).

  1. L199: “which can be explained, probably, by ist morphology” what is “ist”?

We modified the word „ist”, beceause was a mistake, with „amaranth seed” (Line 223).

  1. L210-211: “Bulk density (BD) of amaranth flour particle size values varied between 0.35 210 and 0.58 (g/mL) and presented significant difference (p < 0.05) only regarding particle AM compared to the particles AL and AS.”

We reworded the sentence (Line 232).

  1. L213: “3.2. Physico-chemical composition of wheat-amaranth composite flours” Change “composition” to “properties”. There is no “physical composition” thermology.  This is also to be consistent with the wording in Line 218.   Alternatively, you can change “Physico-chemcial composition of…” to “Chemical composition and color parameters of….”

         Thank you very much for referee. We modiefied “composition” with “properties” at subtitle 3.2 (Line 245).

  1. Table 2 should be proceeded with another table (a new table), showing chemical composition and color parameters of amaranth (before particle size reduction, serve as a control), AL, AM and AL samples. This new table will help support discussion in Line 223-269, and many other places.  It is a pity that the authors overlooked this important aspect.  This missing piece of information is rather important for improving the quality of the manuscript. The rationale is that the particle size effect is magnified by two important properties of flour with difference particle sizes: chemical composition and physical properties (such as color, particle size itself, and water holding capacity, etc.)

               Thank you for your great suggestion, it is of real use for the improvement of the manuscript. We included data on the chemical composition and color parameters for all amaranth flours: integral and their particle sizes. We also pointed out the functional properties of integral amaranth flour. We chose to comprise all the data regarding the chemical composition, functional properties, and color of these flours in Table 1, especially since you said that Table 1 has enough space. I hope you agree with our choice. Then we discussed and highlighted the differences between the chemical composition and physical properties of amaranth flour different particle sizes and integral amaranth flour (Line 177- 244).

  1. L479: Subtitle of 3.7, change to “wheat-amaranth proximate composition flour” to “proximate composition of wheat-amaranth composite flour”

         We modified the subtitle 3.7. from “wheat-amaranth proximate composition flour” to “proximate composition of wheat-amaranth composite flour” (Line 460).

  1. L536-537: “while higher level (15-20%) worsened dough rheology, which resulted also dependent upon flour particle size.”  This sentence is not clear. Need rewording. 

               We reformulated the Conclusions section (Line 506-524).

  1. Furthermore, authors need to add one or two sentences for summarizing how particle size affected dough and bread quality. Since the main sale point of the article is adding particle size as another dimension (in addition to inclusion level), the conclusion should elaborate the effect of particle size as well as its interaction with inclusion level.

We reformulate the Conclusions section, where we highlighted the combined effect of AF addition level and particle size on wheat flour chemical composition, dough rheology, and bread quality (Line 506-524).

We would like to thank to Reviewers for all their comments and suggestions, which have helped us to correct our work and present it in a more acceptable form.

We revised and improved the manuscript according to the editor's suggestions and we resubmit.

Round 2

Reviewer 1 Report

The manuscript has been modified and improved. No other modifications seem necessary.

Author Response

Thank you very much for the opportunity to improve our manuscript to be considered for publishing.

Reviewer 2 Report

THE MANUSCRIPT HAS BEEN IMPROVED AND ALL THE REFEREES' QUERIES SOLVED. THE ENGLISH SHOULD BE REVISED WITH A CAREFUL SPELL CHECK

Author Response

Thank you very much for the opportunity to improve our manuscript to be considered for publishing.

The manuscript was carefully spell-checked.